# Estimating dengue transmission intensity from serological data: A comparative analysis using mixture and catalytic models

**Victoria Cox**[1]*, **Megan O'Driscoll**[2], **Natsuko Imai**[1], **Ari Prayitno**[3], **Sri Rezeki Hadinegoro**[3], **Anne-Frieda Taurel**[4], **Laurent Coudeville**[5], **Ilaria Dorigatti**[1]

**1** MRC Centre for Global Infectious Disease Analysis; and the Abdul Latif Jameel Institute for Disease and Emergency Analytics, School of Public Health, Imperial College London, London, United Kingdom, **2** Department of Genetics, University of Cambridge, Cambridge, United Kingdom, **3** Department of Child Health, Faculty of Medicine Universitas Indonesia, Jakarta, Indonesia, **4** Sanofi Pasteur, Singapore, **5** Sanofi Pasteur, Lyon, France

* v.cox@imperial.ac.uk

## Abstract

### Background

Dengue virus (DENV) infection is a global health concern of increasing magnitude. To target intervention strategies, accurate estimates of the force of infection (FOI) are necessary. Catalytic models have been widely used to estimate DENV FOI and rely on a binary classification of serostatus as seropositive or seronegative, according to pre-defined antibody thresholds. Previous work has demonstrated the use of thresholds can cause serostatus misclassification and biased estimates. In contrast, mixture models do not rely on thresholds and use the full distribution of antibody titres. To date, there has been limited application of mixture models to estimate DENV FOI.

### Methods

We compare the application of mixture models and time-constant and time-varying catalytic models to simulated data and to serological data collected in Vietnam from 2004 to 2009 (N ≥ 2178) and Indonesia in 2014 (N = 3194).

### Results

The simulation study showed larger mean FOI estimate bias from the time-constant and time-varying catalytic models (-0.007 (95% Confidence Interval (CI): -0.069, 0.029) and -0.006 (95% CI -0.095, 0.043)) than from the mixture model (0.001 (95% CI -0.036, 0.065)). Coverage of the true FOI was > 95% for estimates from both the time-varying catalytic and mixture model, however the latter had reduced uncertainty. When applied to real data from Vietnam, the mixture model frequently produced higher FOI and seroprevalence estimates than the catalytic models.

**Data Availability Statement:** The serological data from Vietnam and Indonesia used in this work is proprietary and cannot be publicly shared. Qualified

researchers may request access to the data provider. Further details on Sanofi's data sharing criteria and process for requesting access can be found at: https://www.clinicalstudydatarequest. com. The simulated data can be recreated using code that we have made available in the following GitHub repository: https://github.com/Tori-Cox/Mixture-catalytic-models.

**Funding:** This work was supported by the MRC Centre for Global Infectious Disease Analysis (reference MR/R015600/1), jointly funded by the UK Medical Research Council (MRC) and the UK Foreign, Commonwealth & Development Office (FCDO), under the MRC/FCDO Concordat agreement and is also part of the EDCTP2 programme supported by the European Union. I.D. acknowledges research funding from a Sir Henry Dale Fellowship funded by the Royal Society and Wellcome Trust [grant 213494/Z/18/Z]. V.C. acknowledges funding from the Wellcome Trust [grant 222375/Z/21/Z]. The funders had no role in study design, data collection and analysis, decision to publish, or preparation of the manuscript.

**Competing interests:** The authors have declared that no competing interests exist.

## Conclusions

Our results suggest mixture models represent valid, potentially less biased, alternatives to catalytic models, which could be particularly useful when estimating FOI from data with largely overlapping antibody titre distributions.

### Author summary

Characterising the transmission intensity of dengue virus is essential to inform the implementation of interventions, such as vector control and vaccination, and to better understand the environmental drivers of transmission locally and globally. It is therefore important to understand how methodological differences and model choice may influence the accuracy of estimates of transmission intensity. Using a simulation study, we assessed the performance of catalytic and mixture models to reconstruct the force of infection (FOI) from simulated antibody titre data. Furthermore, we estimated the FOI of dengue virus from antibody titre data collected in Vietnam and Indonesia. The models produced consistent estimates of FOI when they were applied to data with clear separation between the distributions of seronegative and seropositive antibody titres. We observed greater bias in FOI estimates obtained from catalytic models than from mixture models when they were applied to data with high overlap in the bimodal distribution of antibody titres. Our results indicate that mixture models could be preferential to estimate dengue virus FOI when the antibody titre distributions of the seronegative and seropositive components largely overlap.

## Introduction

Dengue fever is caused by infection with one or more of four antigenically distinct serotypes of dengue virus (DENV1-4), a *Flavivirus* carried by *Aedes* mosquitoes [1,2]. DENV infects approximately 105 million people each year [3], primarily in tropical and sub-tropical regions. The geographical range of DENV is increasing [1,4,5] and it is expected that the spread of dengue will be influenced by rising global temperatures and increasing urbanisation [1,6]. Intervention measures to date rely essentially on vector control due to the absence of antiviral treatment, challenges in the use of the first licensed dengue vaccine for widespread dengue prevention and control [7], as well as in the use of rapid diagnostic tests for screening [8]. The current and expected future burden of dengue on health-systems is therefore high, demonstrating a continuing need for increased understanding of DENV transmission.

Estimating epidemiological parameters such as the force of infection (FOI, the per capita rate at which a susceptible person is infected) and population seroprevalence (the proportion of people in a population exposed to a virus, as determined by the detection of antibodies in the blood) allow us to gain insights into the subsets of populations most at risk of infection and disease [9], to assess the predicted impact of an intervention strategy [10], and to inform public health policy [11,12].

Both the FOI and seroprevalence can be estimated using mathematical models calibrated to age-stratified serological data measuring IgG antibody levels (also called titres) from blood samples. IgG titres are obtained using Enzyme-Linked Immunosorbent Assays (ELISAs) and are often classified into qualitative, binary test results (seropositive or seronegative) based on the manufacturer's threshold.

Catalytic models, first proposed in 1934, estimate disease FOI from age-stratified serological or case notification data [13]. In these models, large rates of increase in seroprevalence between individuals who are age *a* versus age *a+1* are explained by high age-specific FOI (assuming the FOI is constant in time) or high time-specific FOI experienced by individuals of all ages during the period *a* to *a+1* years ago [14]. Catalytic models have been used extensively for measles [15], rubella [16], Hepatitis A [17], Chagas disease [18], and DENV [12,14,19–21]. Whilst commonly used, previous work suggests that catalytic models risk generating biased estimates due to data-loss and/or misclassification [22–24]. For example, samples with titres greater than the seronegative threshold but lower than the seropositive threshold are classified as 'equivocal' and discarded from the analysis. Furthermore, titre levels of seropositive individuals in a given population may be affected by factors including host response, the degree of exposure to the pathogen and infection timing, which could lead to misclassification.

Mixture models are flexible statistical models that can be applied to continuous data from different clusters or populations, called components. Mixture models can therefore be applied to the absolute antibody titre values in serology datasets, rather than to the counts of titres in each of two classes (seropositive/seronegative) as is necessary for catalytic models [22]. The components' distributions and their defining parameters (e.g., the mean titre of each component distribution) are inferred from a fitted mixture model which is used to estimate the FOI and population seroprevalence [22,25]. To date, mixture models have been applied to serological data to estimate the seroprevalence of infectious diseases such as parvovirus B19 and rubella in England [26,27], human papillomavirus in the Netherlands [23], measles in Italy [28], and a selection of arboviruses inlcuding DENV in Zambia [29]. In addition, mixture models have been used to develop frameworks capable of distinguishing between primary and post-primary DENV infections [30,31], and recent and historical influenza A infections [32]. Recently, DENV FOI was estimated using catalytic and mixture models applied to serological data collected in three locations in Vietnam (N > = 266) and in Chennai, India (N = 799) [31]. In this study, the estimates from mixture models were deemed more robust than those from binary catalytic models [31].

Here, we implement a simulation study to assess the ability of mixture and catalytic models to reconstruct the FOI value used for simulating the data. Furthermore, we add to the growing body of evidence exploring the use of mixture models by presenting a comparitative analysis of the DENV trasmission intensity estimates obtained from mixture and catalytic models applied to age-stratified serological datasets from Vietnam (N ≥ 2178, for years 2004–2009) and Indonesia (N = 3194, for 2014).

## Methods

### Ethics statement

Ethical approval for the secondary analysis of the age-stratified seroprevalence datasets was granted by the Imperial College Research Ethics Committee (Approval Reference 21IC7066).

### Data

**Age-stratified seroprevalence data.** DENV IgG data were collected in Long Xuyen, Vietnam, during a prospective epidemiological study that was conducted to assess the suitability of the area for future CYD-TDV vaccine efficacy trials, as described previously [33]. Samples were collected from children under 11 years old in 2004 and then from children under 15 years old during September to February in 2004–2005, 2005–2006, 2006–2007, 2007–2008 and 2008–2009 (Datasets A-1 to A-6, Table 1). The titres were measured using in-house ELISA assays (Arbovirus Laboratory of Pasteur Institute, Ho Chi Minh City).

**Table 1. Description of the datasets used in the analyses.** Summary statistics including notation, region, the assay used, the year of testing, the age range of the children participating to the study and the sample sizes.

| Dataset | Region | Assay type | Year | Age Range | Sample Size |
|---------|--------|------------|------|-----------|-------------|
| A-1 | Long Xuyen, Vietnam | IgG ELISA | 2004 | 3–10 | 2,178 |
| A-2 | Long Xuyen, Vietnam | IgG ELISA | 2004–2005 | 3–13 | 3,681 |
| A-3 | Long Xuyen, Vietnam | IgG ELISA | 2005–2006 | 3–14 | 3,727 |
| A-4 | Long Xuyen, Vietnam | IgG ELISA | 2006–2007 | 3–14 | 3,651 |
| A-5 | Long Xuyen, Vietnam | IgG ELISA | 2007–2008 | 4–14 | 2,959 |
| A-6 | Long Xuyen, Vietnam | IgG ELISA | 2008–2009 | 5–14 | 2,249 |
| B | 30 urban subdistricts, Indonesia | IgG ELISA | 2014 | 1–18 | 3,194 |
| C | Simulated data | Simulated | N/A | 1–18 | 3,194 |

DENV IgG data from 30 urban subdistricts in Jakarta, Indonesia were collected from 3,194 children under 18 years old as part of a cross-sectional seroprevalence survey in 2014 [34] (Dataset B). Given the small spatial scale of the range of data collection, we did not account for spatial differences when modelling. IgG titres were measured using the commercial Panbio Dengue IgG Indirect ELISA kit.

**Simulated datasets.** We simulated 540 antibody titre datasets (Dataset C), with the same age-distribution and sample size as the Indonesian seroprevalence survey data (Dataset B). For each simulation the distributions used for sampling seronegative and seropositive log(titres + 1) were selected from a normal, gamma or Weibull distribution. This gave 9 possible distribution pairs for seronegative and seropositive log(titre + 1) values, and we generated an equal number of simulations (N = 60) for each combination. Normal, gamma and Weibull distributions were chosen based on preliminary work on our antibody titre datasets showing that these mistures were most frequently selected among a wider set of distributions. Parameter values were randomly drawn from uniform distributions with limits as shown in S1 Table. The serostatus of each individual was drawn from a Bernoulli distribution with probability $1-e^{-\lambda a}$, where $a$ is the age of the individual and $\lambda$ is the FOI (which is assumed to be constant with age and time), and therefore $\lambda a$ represents the cumulative FOI experienced by individuals over their lifetime. Log(titre + 1) values for each individual were subsequently randomly drawn from the respective component distributions. The analysis was conducted in the statistical programming language R [35].

## Catalytic model

**Data preparation.** Catalytic models rely on data that are binarily classified as seropositive or seronegative. For Datasets A-1 to A-6, a background/control titre (t) was measured for each assay. An individual titre was classified as seronegative if $\leq t$ and seropositive otherwise. For Dataset B, samples with titres $\leq 9$ PanBio units were classified as seronegative and $\geq 11$ as seropositive. Titres $>9$ and $<11$ were discarded (28 out of 3,194 samples). For simulated Dataset C, titres were classified as seronegative if they were $\leq X$ and seropositive if they were $\geq Y$. X and Y are thresholds that were optimised using the 'true' simulated serostatuses: the *optim* function in R, using the Nelder Mead algorithm, was used to calulate the X and Y values per simulated dataset resulting in the fewest titre misclassifications. The optimisation process occassionally failed to estimate realistic classification thresholds (X < 25% quantile of seronegative titres or Y > 75% quantile of seropositive titres) and these simulations were excluded from analysis (N = 31). For the remaining 509 simulations, titres $>X$ and $<Y$ were classified as equivocal and discarded (mean = 2 out of 3,194 samples, median = 1, interquartile range

(IQR) = 2, range = 0 to 30). The mean percentage misclassification error rate of titres across the simulations was 6.6% (median = 2.8%, IQR = 9.8%, range = 0% to 41.9%).

**Parameter estimation.** We used a catalytic model as previously described [14,36]. The yearly FOI, i.e., the per capita rate of infection experienced by individuals in a given year $X-i$, where $X$ is the year the serosurvey was conducted, is assumed to be either constant in time (constant across the years) or time-varying (piecewise constant across the years). When we assumed a time-varying FOI, the number of FOI estimates is equal to the number of single year age groups available in the datasets (maximum age group $A$ –minimum age group $M$). The proportion of seropositive individuals in age group $a$ during year $X$ ($\pi_{a,X}$), was estimated as in Eq 1. Here, the yearly FOI ($\lambda_{X-i}$) is summed over the lifetime of the individuals in age group $a$ to give the cumulative FOI experienced by the individuals in this age group. If the minimum age group $M$ does not equal 1, then we estimated an average FOI for the $M$ years without age-specific data ($X-M$ to $X$) denoted $\lambda_{X-M}$.

$$\pi_{a,X} = 1 - exp(-\sum_{i=0}^{a}\lambda_{X-i}). \tag{1}$$

When we assume a time-constant FOI over the whole study period, Eq 1 can be expressed as shown in Eq 2:

$$\pi_{a,X} = 1 - exp(-\lambda \cdot a). \tag{2}$$

A binomial log-likelihood was assumed for the FOI (Eq 3), where $N_a$ is the total number of individuals and $P_a$ is the number of seropositive individuals in age group $a$ during year $X$ [19]. The *optim* function in R was used to find the maximum likelihood estimate of the FOI using Eq 3.

$$LnL(\lambda_{X-i}) = \Sigma_a[(N_{a,X} - P_{a,X})(log(1 - \pi_{a,X}) + P_{a,X}(log(\pi_{a,X})]. \tag{3}$$

When we assumed a time-varying FOI, the $\lambda_{X-i}$ values were averaged to produce a mean FOI experienced over the years in the study period ($A-M$) (Eq 4) which was compared to the FOI estimated by the time-constant catalytic model and the mixture model.

$$\lambda = \frac{\sum_{i=0}^{A}\lambda_{X-i}}{A-M}. \tag{4}$$

We estimated 95% Confidence Intervals (CI) using a bootstrap method, where the titre data were sampled with replacement and the age-stratified proportion of seropositive indivuals was calculated, 500 times. The 95% CI was given by the 2.5% to 97.5% quantiles of the estimates from the catalytic models applied to the bootstrap samples.

## Mixture model

**Applying the mixture models to the titre distributions.** Mixture models were applied to the bimodal distribution of individual antibody titres as described in Bollaerts *et al.*, 2012 and Hens *et al.*, 2012 [22,25]. All individual antibody titre measurements were used in each dataset, which differs from the data used for the catalytic model where equivocal titres were discarded and titre measurements were classified as either seropositive or seronegative. The mixture model defines the distribution ($z$) of the log(titres + 1) as a mixture of two distinct distributions: one for susceptible individuals (seronegative, $z_s$) and one for individuals who have been previously infected (seropositive, $z_I$). The two-component mixture model is represented by:

$$f(z|z_s, z_I, a, X) = (1 - \pi_{a,X})f_s(z_s|\mu_s, \sigma_s) + \pi_{a,X}f_I(z_I|\mu_I, \sigma_I), \tag{5}$$

where $f_s$ and $f_I$ represent the probability density function of the seronegative and seropositive components, respectively, and where $\mu$ and $\sigma$ represent the mean and standard deviation of each component, and $\pi_{a,X}$ represents the age-specific seroprevalence during year $X$, when the serosurvey was conducted.

The *mixdist* R package was used to fit the mixture models to the titre data by maximum likelihood using an Expectation Maximisation (EM) algorithm [37]. The package was adapted to allow fitting of different distributions for the seronegative and seropositive titre components: normal, gamma and Weibull distributions, giving 9 possible combinations. The best fitting mixture was chosen using the Akaike Information Criterion (AIC). For Dataset C, the estimated means ($\mu_s$ and $\mu_I$) and standard deviations ($\sigma_S$ and $\sigma_I$) for the seronegative and seropositive components of the best mixture were compared against the true parameter values used for simulating the data. We explored multiple parameterisations, including fixing the standard deviation of the two mixture components. For the Vietnamese datasets, we optimised the model having constrained the standard deviation of the seropositive component to multiple different values (for Dataset A-4 we set $\sigma_I$ equal to all values from 0.02 to 0.08 in steps of 0.01, for the other five Datasets we set $\sigma_I$ equal to all values from 0.05, to 0.15 in steps of 0.01). For the Indonesian dataset (Dataset B) the standard deviations of both components were constrained ($\sigma_S$ was set equal to 0.10 to 0.15 in steps of 0.001, and $\sigma_I$ was set equal to 0.15 to 0.3 in steps of 0.05).

**Parameter estimation.** The relationship between the age-dependent mean log(titre + 1) ($\mu_{a,X}$), the age-specific seroprevalence ($\pi_{a,X}$) and the means of the mixture components ($\mu_s$ and $\mu_I$) is described in Eq 6. We estimated $\mu_{a,X}$ by least-squares regression using a monotonically increasing P spline [22,25,38] using the *mpspline.fit* function from the *serostat* R package [39]. Equally spaced cubic polynomial segments (degree = 3) made up the spline. The optimal smoothing parameter ($\alpha$) and number of segments (knots) were determined using the Bayesian Information Criterion, having explored combinations of $\alpha$ values (set equal to 0.001, 0.01, 0.1, 0.5, 1, 5, 10, 50, 100, 500) and knots (set equal to values in the sequence: 5 to the maximum number of x-axis age categories, step size = 1). The seroprevalence was calculated using Eq 7.

$$\mu_{a,X} = (1 - \pi_{a,X})\mu_s + \pi_{a,X}\mu_I. \tag{6}$$

$$\pi_{a,X} = \frac{\mu_{a,X} - \mu_s}{\mu_I - \mu_s}. \tag{7}$$

The time-varying FOI was derived from the age-specific seroprevalence as described in Eq 8 [22], where the rate of change in the seroprevalence between two sequential age groups ($a-1$ and $a$) is divided by the proportion of seronegative individuals in age group $a$, to give the FOI experienced in the year $X-a$ ($\lambda_{X-a}$). Eq 8 can in turn be expressed as a function of the underlying antibody titre distribution as shown in Eq 9, where $\mu'_{a,X}$ represents the derivative of the age-specific mean log(titre + 1). The $\mu'_{a,X}$ terms were calculated by taking the gradient of the fitted $\mu_{a,X}$ spline at each age group $a$. The time-varying FOI can be averaged across the years in the study period to give the total FOI $\lambda$ (Eq 4).

$$\lambda_{X-a} = \frac{\pi'_{a,X}}{1 - \pi_{a,X}}. \tag{8}$$

$$\lambda_{X-a} = \frac{\mu'_{a,X}}{\mu_I - \mu_{a,X}}. \tag{9}$$

The 95% CI around the FOI and seroprevalence estimates were calculated using a boostrap

method, where the titre data were sampled with replacement 5000 times. The 95% CI were given by the 2.5% to 97.5% quantiles of the estimates from the bootstrap samples.

Bias in the mixture and catalytic model estimates of FOI and seroprevalence for Dataset C was calculated as the estimated value minus the true simulated value of the parameter. Uncertainty was calculated as the width of the 95% CIs around the parameter estimates. Coverage was calculated as the percentage of simulations where the estimated 95% CIs contained the true parameter value. Code for the simulation study analysis is available at: https://github.com/Tori-Cox/Mixture-catalytic-models.

## Results

### Simulated data

The mixture model identified the correct distributions used to simulate both seropositive and seronegative titres in 76.1% (411/540) of simulations, one of the two distributions in 22.2% (120/540) of simulations and did not correctly identify either distribution in 1.7% (9/540) of the simulations. Whether the distributions were gamma, normal or Weibull did not influence the ability of the mixture model to correctly identify them (S2 Table). The estimated 95% CIs contained the true parameter values used to simulate the data in 88.1% (476/540), 86.9% (469/540), 86.5% (467/540) and 89.4% (483/540) of simulations for $\mu_s$, $\mu_I$, $\sigma_S$ and $\sigma_I$, respectively (S1A Fig). Simulations where the seronegative titre distribution was Weibull distributed were over-represented in the simulations which produced outlying estimates of $\mu_s$, $\mu_I$, $\sigma_S$ and $\sigma_I$ (S1B Fig).

The mixture model coverage for the FOI was 95% of the total simulations (513/540) and 95% (485/509) of the simulations included in the catalytic model analysis, and for the seroprevalence was 88% (475/540) and 89% (451/509) respectively. The time-varying catalytic model coverage for the FOI was 96.7% (492/509) and for the seroprevalence was 78.8% (401/509). The time-constant catalytic model coverage for the FOI and seroprevalence was 38.9% (198/509) and 55.0% (280/509). It should be noted that the time-varying catalytic model produced wider CIs compared to when assuming a time-constant FOI or when using a mixture model (Fig 1). Average bias in the FOI estimates (0.001 (95% CI -0.036, 0.065), -0.007 (95% CI -0.069, 0.029) and -0.007 (95% CI -0.095, 0.043) for the mixture, time-constant and time-varying catalytic models, respectively) and the seropreavalence estimates (-0.003 (95% CI -0.144, 0.108), -0.007 (95% CI -0.244, 0.087) and -0.005 (95% CI -0.241, 0.100)) was smaller for the mixture model estimates (Fig 1). The increased negative bias in the catalytic model estimates compared to the mixture model estimates demonstrates that the catalytic models are more prone to underestimation of FOI and seroprevalence (Figs 1 and 2). High antibody titre misclassification error rates were positively associated with increased bias in the parameter estimates from the catalytic models (S3 Fig). As expected, model performance was improved when we fitted the catalytic models to the simulated 'true serostatus' (i.e., without classifying the titres using optimised thresholds): the coverage of the FOI was 99% (536/540) (95% CI: 98%, 100%) and 42% (228/540) (95% CI: 38%, 47%) for the time-varying and time-constant FOI catalytic models respectively, and the coverage of the seroprevalence was 100% for both models. The average bias in the FOI estimates was 0.007 (95% CI -0.020, 0.056) and -0.005 (95% CI -0.016, -0.477), and in the seroprevalence estimates was 0.000 (95% CI -0.001, 0.001) and 0.002 (95% CI -0.001, 0.001) for the time-varying and time-constant FOI catalytic models respectively.

### Long Xuyen, Vietnam data

When we applied the mixture model (Fig 3) to the data from Long Xuyen, Vietnam, the total population-level seroprevalence estimates ranged from 0.163 (95% CI 0.138–0.188) in 2004 to

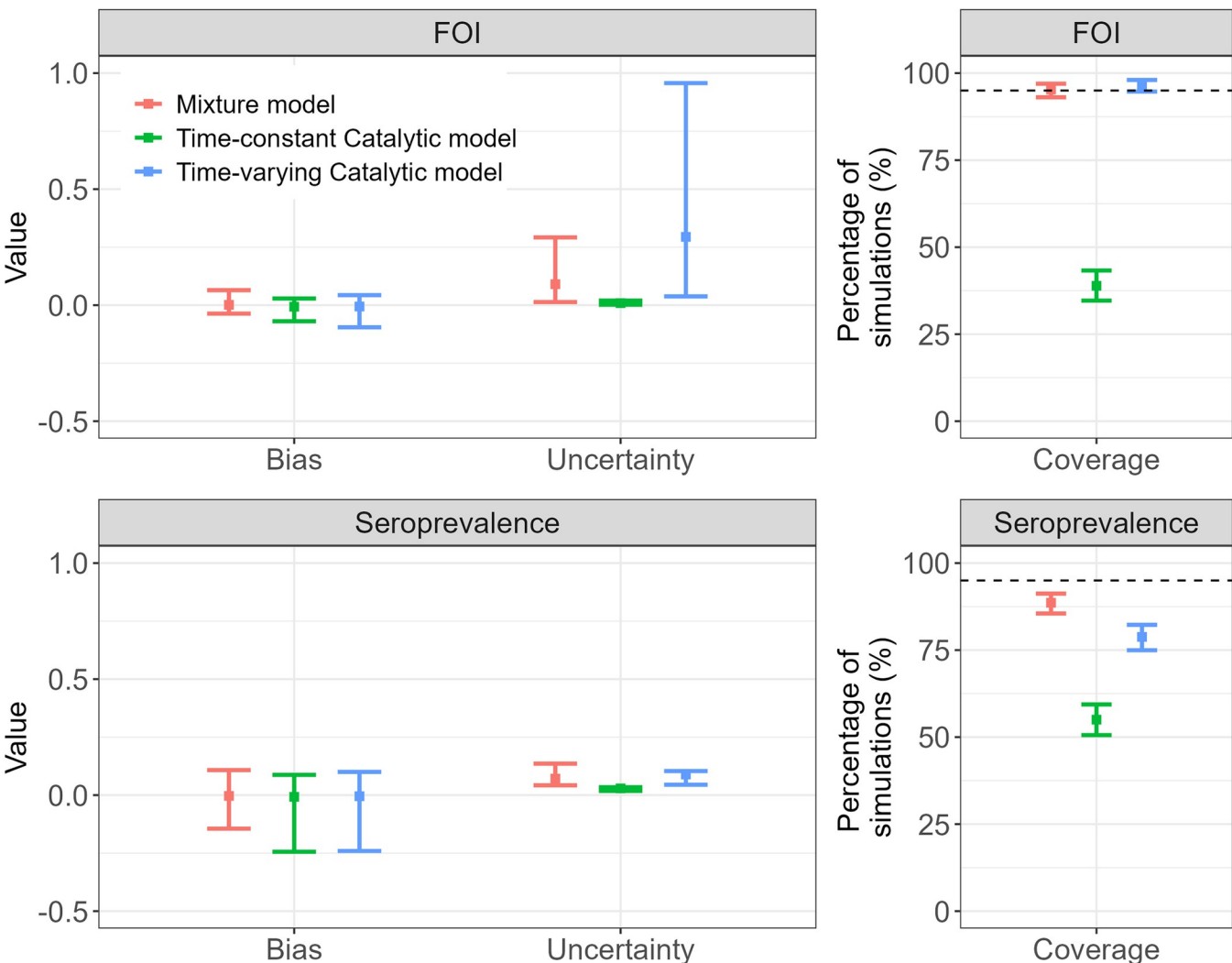

**Fig 1. Bias, coverage, and degree of uncertainty for seroprevalence and force of infection (FOI) estimates using catalytic and mixture models fitted to simulated datasets (Dataset C).** Bias is calculated as the estimated parameter value minus the true parameter value. Uncertainty is the width of the 95% Confidence intervals (CIs) around the central estimates, calculated using the bootstrap method. The coverage is the percentage of simulations where the estimated CIs contained the true values. The dashed line at 95% shows the threshold for the ideal coverage. For the bias and the uncertainty, the mean and 95% CI across the 509 simulations are given. For the coverage, the 95% exact binomial CI are given.

0.376 (95% CI 0.249–0.403) in 2005–2006. The seroprevalence estimates from the time-constant and time-varying catalytic models were consistent with each other, with the latter ranging from 0.189 (95% CI 0.163–0.217) in 2006–2007 to 0.299 (95% CI 0.262–0.337) in 2008–2009. The seroprevalence estimates from all three models were consistent (as determined by the 95% CIs) for 4 out of 6 datasets (Fig 4, S3 Table). The general trend in the age-specific seroprevalence estimates, specifically for Datasets A-2:A-5, differed significantly between the mixture model and the catalytic models, with the mixture model estimating higher seroprevalence at the older ages (Fig 5).

The average FOI estimated by the mixture model ranged from 0.026 (95% CI 0.019–0.033) for the period 1993 to 2004, to 0.099 (95% CI 0.077–0.124) for 1990 to 2005. The average FOI estimated by the time-varying catalytic model ranged from 0.024 (95% CI 0.007–0.058) for the period 1991 to 2007, to 0.050 (95% CI 0.001–0.118) for 1990 to 2005. The FOI estimates from

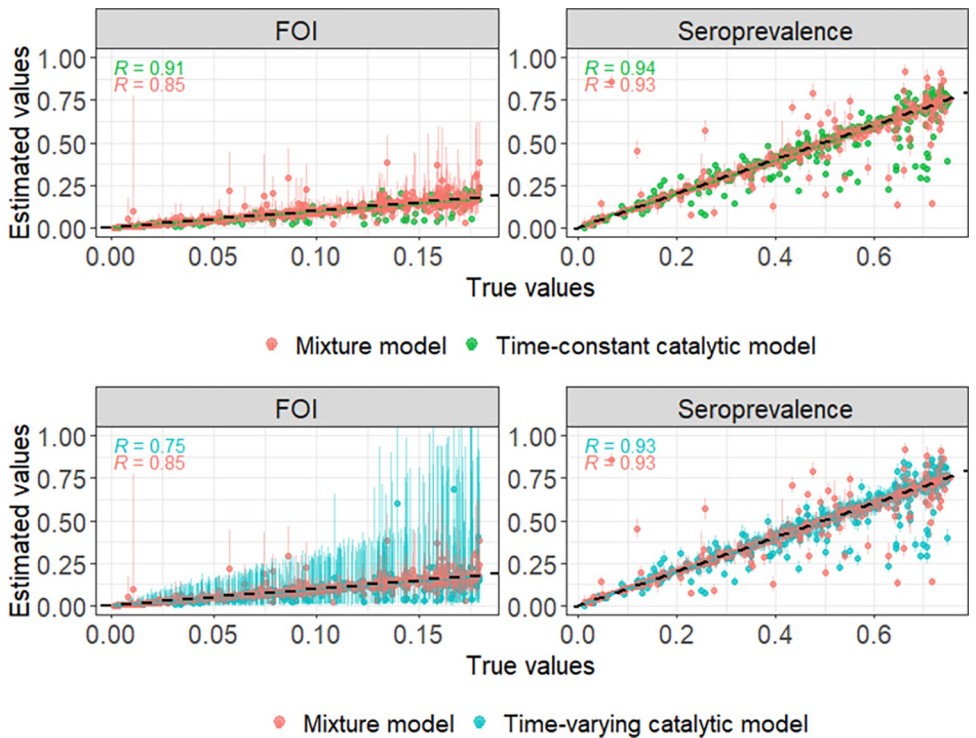

**Fig 2. True versus estimated seroprevalence and force of infection (FOI) values from the mixture and catalytic models fitted to the simulated datasets (Dataset C).** The catalytic model was run under the assumption that the FOI was time-constant or time-varying. The 95% Confidence Intervals for the estimated values were calculated using a bootstrap method and are shown here as error bars; the point denotes the central estimate. The Pearson's correlation coefficients (*R*) are shown. The dashed line represents the line y = x and shows where points would be located in a scenario with zero bias in the estimated values.

the mixture model versus the time-varying catalytic model were consistent for 6 out of 6 datasets, and versus the time-constant catalytic model they were consistent for 3 out of 6 datasets (Fig 4, S3 Table). There is a higher degree of uncertainty around the estimates from the catalytic model when assuming a time-varying FOI compared to the time-constant FOI assumption (Fig 4). We observe greater differences in the estimates from each model when comparing the year-specific FOI as opposed to the averaged total FOI (S4 Fig).

## Indonesian data

The mixture and catalytic models fitted to the Indonesian data produced consistent FOI, total population seroprevalence and age-specific seroprevalence estimates. The FOI for the period 1996 to 2014 was estimated at 0.154 (95% CI: 0.106–0.213), 0.143 (95% CI 0.136–0.150) and 0.164 (95% CI 0.022–0.814), and the seroprevalence in 2014 was estimated at 0.718 (95% CI 0.694–0.741), 0.700 (95% CI 0.686–0.714) and 0.700 (95% CI 0.655–0.743) by the mixture model and the time-constant and time-varying catalytic models, respectively (Fig 4, S3 Table).

## Discussion

In this analysis, we explored the accuracy and bias of FOI and seroprevalence estimates obtained from mixture and catalytic models applied to serological data. The catalytic models were applied assuming a time-constant or time-varying FOI. We performed a simulation study to compare the performance of each model with known parameter values used to

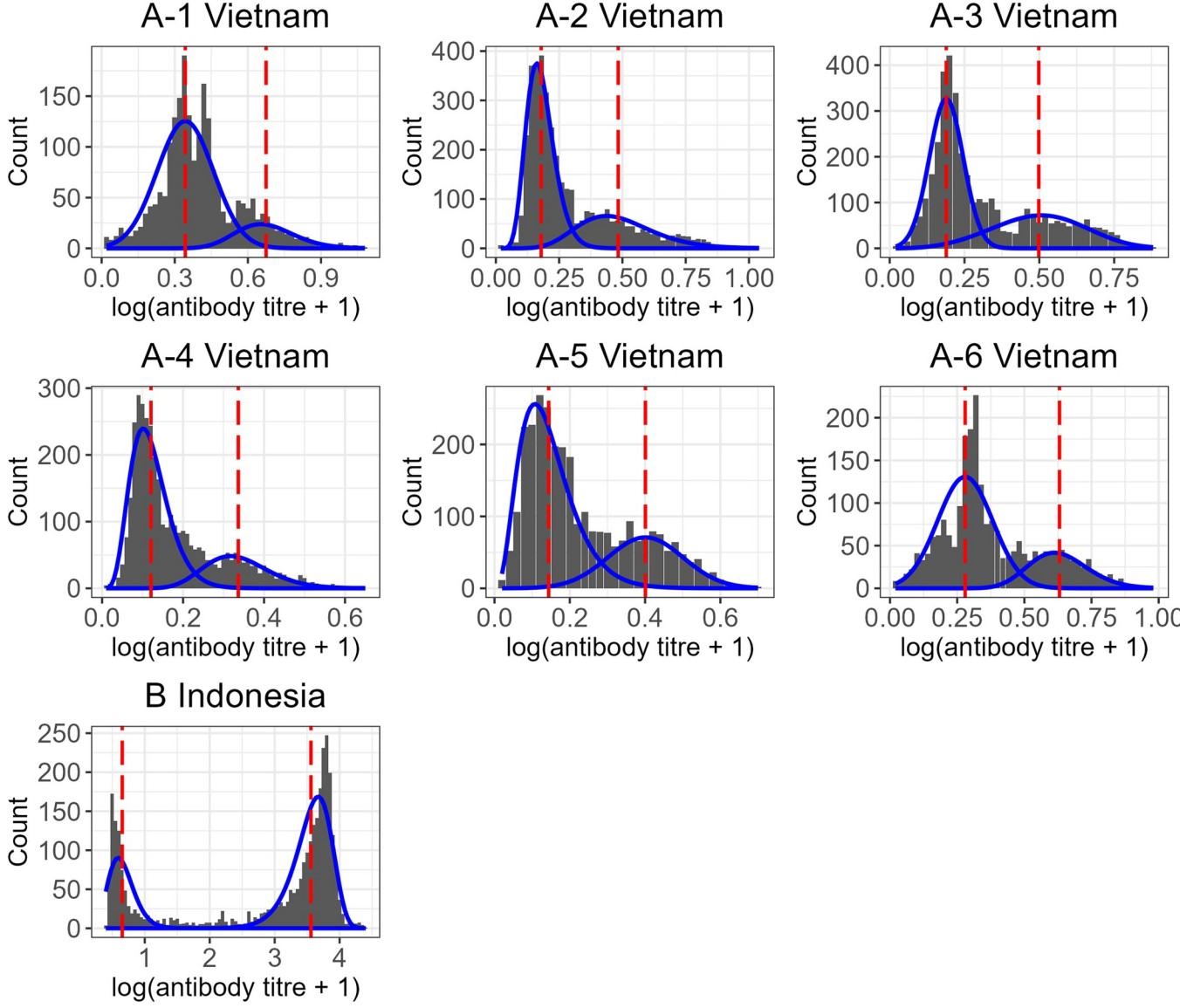

**Fig 3. Mixture model fitted to the Vietnamese (A1:A6) and Indonesian (B) datasets.** The distribution of log(titre+1) is shown in dark grey, the fitted mixture model is shown in blue, and the red dashed lines represent the mean antibody titre of each component of the fitted mixture model ($\mu_s$ and $\mu_I$ for the seronegative and seropositive components respectively). Note that the y-axis limits differ for each panel.

generate the simulated data, and we observed significantly greater accuracy in FOI and seroprevalence estimates from the mixture and time-varying catalytic models than time-constant catalytic models. We observed reduced bias and uncertainty in estimates from the mixture compared to the time-varying catalytic model.

In our simulation study, larger bias in the catalytic model estimates of FOI and seroprevalence (Figs 1 and 2), was associated with increased serostatus misclassification (S3 Fig). Serostatus misclassification occurred more often in simulations where the difference between the mean log(titre + 1) for the susceptible/seronegative component and the mean log(titre + 1) for the infected/seropositive component was lower (S2 Fig), indicating greater overlap between the distributions of the two components. Our results are consistent with previous work which showed greater bias in seroprevalence estimates using methods which employ cut-off

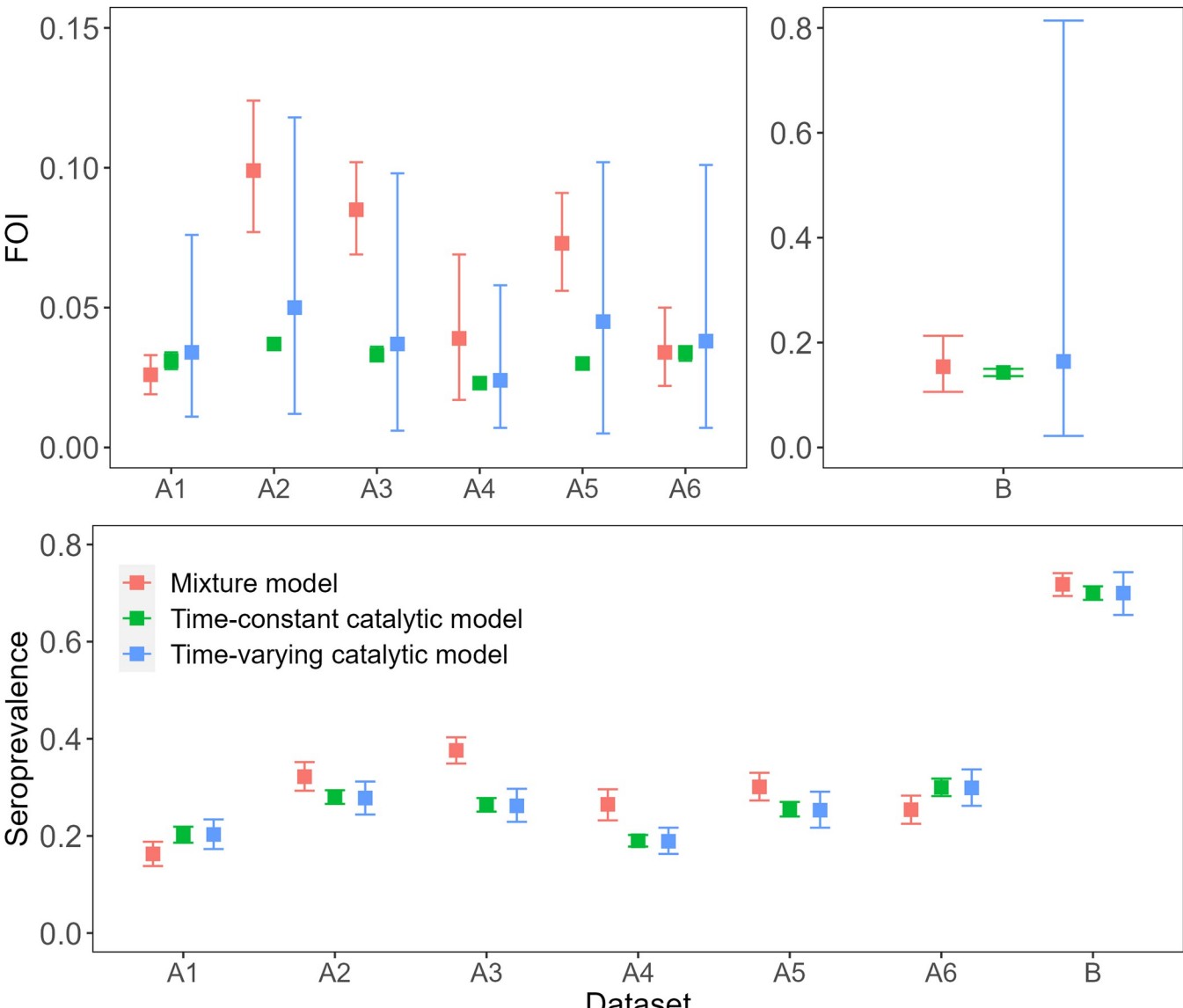

**Fig 4. Force of infection (FOI) and total population level seroprevalence (SP) estimates from the mixture model and the catalytic models fitted to the observed data.** The catalytic model was run under the time-constant and time-varying FOI assumption. The 95% Confidence Intervals (CI) which were calculated by bootstrapping for all models are given as error bars. Note that the y-axis limits differ for each panel.

thresholds to classify simulated antibody data as opposed to mixture models, when there was high overlap in the underlying components [24].

Differences in the degree of overlap between components in real serological datasets are likely impacted by many factors, including differences in the ELISA tests used to measure antibody titres, the age groups sampled and the underlying age structure of the population as well as the transmission setting and spatiotemporal heterogeneities in the risk of infection at the local scale. In datasets where there is clear separation in the bimodal distribution of antibody titres, catalytic and mixture models are expected to produce more similar estimates of FOI and seroprevalence as fewer samples are misclassified during the binary classification of the data needed to calibrate catalytic models [22,24]. This is consistent with the results from our simulation study and with the reduced variability we observe in our FOI and seroprevalence

estimates from each model when they were applied to serological data from Indonesia compared to Vietnam, where the former had higher separation of titre distributions (Figs 3 and 4, S3 Table).

The estimates for Indonesia from each model were consistent with each other and with previously published FOI estimates from catalytic models fitted to case-notification data from 2008–2017 in Jakarta, Indonesia (0.130, 95% CI: 0.129–0.131) [12], and seroprevalence estimates from time-constant catalytic models applied to the same serology dataset (Dataset B) [34,40]. Our results show that the mixture and catalytic models do not significantly differ in their FOI and seroprevalence estimates in this setting. In contrast, the mixture model applied to the six datasets from Vietnam produced more variable estimates (FOI range = 0.026–0.099, seroprevalence range = 0.16.3–0.376) than the catalytic models (FOI range = 0.023–0.037 and 0.024–0.050, seroprevalence range = 0.190–0.300 and 0.189–0.299 under the assumption of a time-constant or time-varying FOI respectively). The variance was even greater in the age-specific seroprevalence and yearly FOI estimates (Figs 5 and S4). As expected, the time-varying catalytic model and the mixture model (which implicitly models FOI as time-varying), were better able to capture the age-specific seroprevalence than the time-constant catalytic model. The estimates from the mixture model tended to exceed those obtained from the catalytic models (Fig 4, S3 Table). Given the greater negative bias observed for the catalytic models in our simulation study, we expect the higher mixture model estimates to be more accurate for the Vietnamese setting. Lam *et al.*, similarly observed higher FOI estimates when applying mixture models compared to catalytic models to serological data from Vietnam, for example 0.12 (95% CI: 0.11–0.14) compared to 0.07 (95% CI: 0.06–0.09), in Ho Chi Minh City [31].

A major advantage of the mixture model is the comparative ease with which it can be applied to serological data to estimate transmission intensity without the need to use thresholds to process the data. Furthermore, to generate robust estimates, there are fewer data requirements for mixture models than for catalytic models: in the former, the data are pooled, and age is used only to calculate the age-specific mean log(titre + 1) using a spline, meaning that there are no constraints on the number of participants per age category. However, it is important to consider the bias that will be introduced if the mixture distributions fit the titre data poorly [24]. In this study we accounted for this by using an information criterion to select the best fitting models from a range of options. In the future we will explore implementing the models in a Bayesian framework [22] which would allow us to perform posterior predictive checks to more robustly assess model fit. It would also be interesting to explore the FOI estimates obtained when applying a mixture model with more than two mixture distributions, which may better account for the complex immunity profiles observed in areas where multiple DENV serotypes circulate. For example, Biggs *et al.* and Lam *et al.* fit three-component mixture distributions to DENV antibody titre data in the Philippines and Vietnam respectively, to develop frameworks capable of distinguishing between post and primary DENV infection [30,31] by specifying mixture components for seronegative, seropositive with a primary infection and seropositive with post-primary infections.

In summary, our results suggest that mixture models represent a good alternative to catalytic models to quantify DENV time-varying FOI and seroprevalence from age-stratified serological data, with potentially less bias and less uncertainty. They may be particularly useful when estimating FOI from data where there is high overlap between the component distributions, where the risk of serostatus misclassification and bias introduction when using cut-off threshold methods is greater (S2 and S3 Figs).

We have provided code to run the simulation study to encourage further exploration and comparison of the different methodologies. Critically, further investigation of the use of mixture models depends on the availability of raw antibody titre data. For these reasons, we would

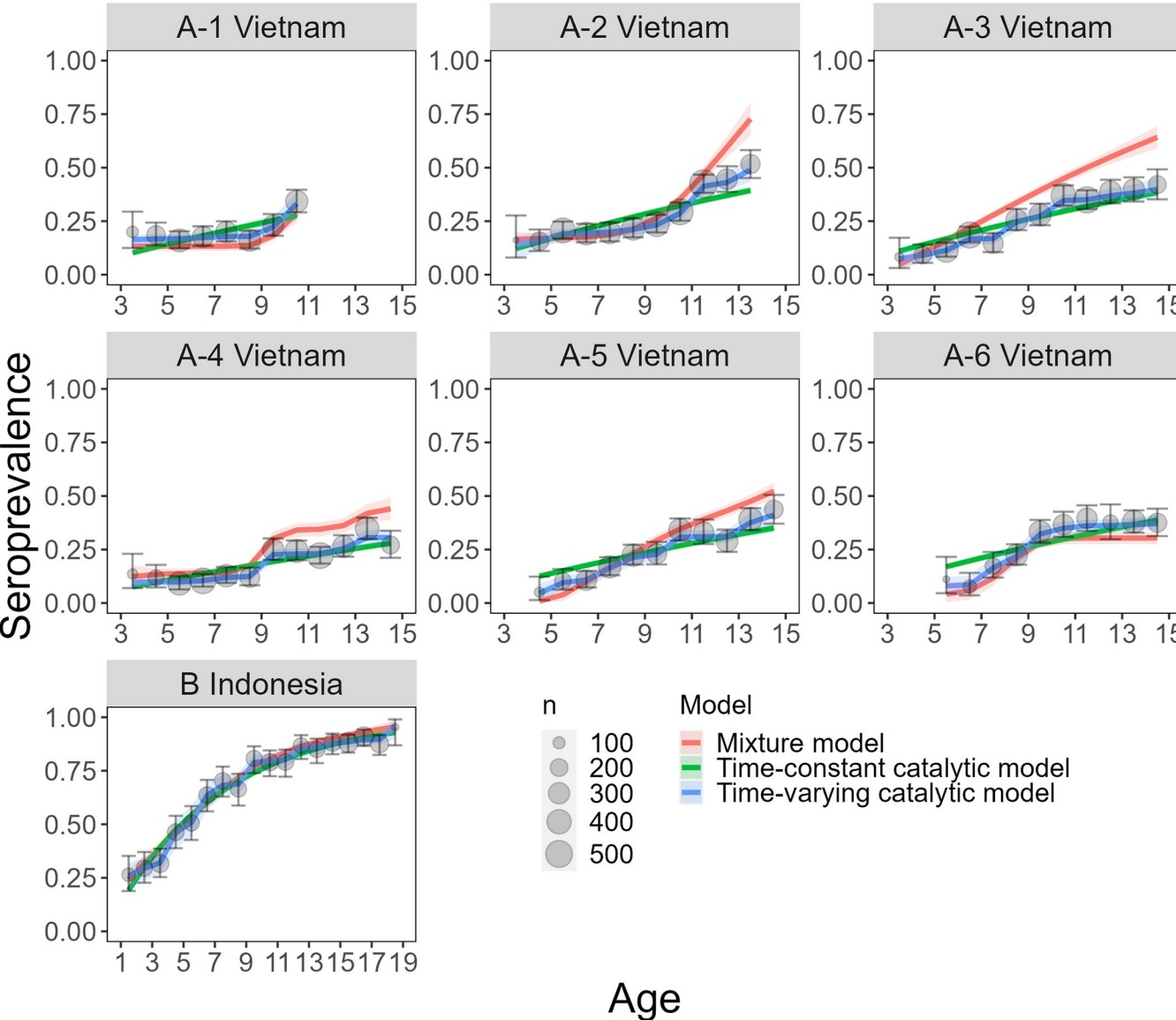

**Fig 5. Age-specific seroprevalence estimates for the IgG data from Vietnam (Dataset A1:A6) and from Indonesia (Dataset B).** Mixture model estimates are in orange, catalytic model estimates are in green and blue when applied under the assumption that the FOI is time-constant or time-varying respectively. Shading represents the 95% Confidence Intervals (CI). The grey points show the observed seroprevalence per age group calculated from the binarily classified IgG data (seropositive individuals / tested individuals), with error bars indicating the 95% exact binomial CIs. The seroprevalence data and model estimates are overlayed for the purpose of comparison. However, it is important to note that the mixture model was not fitted to the data (grey points), as the former does not depend on the titre classification. The size of the grey data points represents the number of individuals tested in each age group.

encourage current and future seroprevalence studies on DENV, as well as other infectious diseases, to publish anonymised individual-level antibody titre data where it is possible to do so.

## Supporting information

**S1 Table. Parameter values used for generating 540 simulated datasets.**
(DOCX)

**S2 Table. Number of simulations where the mixture model correctly specified the distributions of the seronegative and/or seropositive component of the simulated antibody titre**

**datasets (Dataset C).** Here, n represents the number of simulated datasets out of 540.
(DOCX)

**S3 Table. Force of infection (FOI) and total population level seroprevalence (SP) estimates from the mixture model and the catalytic models fitted to the observed data.** The observed data is serology data collected in Vietnam (Datasets A-1:A-6) and Indonesia (Dataset B). 95% Confidence Intervals (CI) were calculated by the bootstrap method.
(DOCX)

**S1 Fig.** (A) True versus estimated parameter values from the mixture model fitted to the simulated datasets (Dataset C). The estimated parameters are the mean log(titre + 1) value of the seronegative/ susceptible (S) and seropositive/infected (I) components (µs and µI respectively) and the corresponding standard deviations (σs and σI). Red indicates the estimates where the true parameter value was not captured by the estimates (i.e., the 95% Confidence Interval of the estimate did not contain the true value). Note that the axes limits differ for each panel. (B) The percentage of parameter outliers after fitting the mixture model to Dataset C, per seronegative and seropositive titre family distributions. The percentage of the total number of outliers of µs, µI, σS and σI (red in panel A) per distribution combination on the x-axis, where the two letters represent the seronegative (first letter) and the seropositive (second letter) distribution pair (N = normal, G = gamma and W = Weibull).
(TIF)

**S2 Fig. Association between the true component mean titre values in Dataset C versus the serostatus misclassification error.** The x-axis shows the difference between the true mean log (titre + 1) value of the seronegative ($\mu_s$) and the seropositive component ($\mu_I$) for each realisation over 509 simulated datasets. The titres are classified as seropositive or seronegative using realisation-specific optimised titre thresholds. The loess regression line and corresponding 95% Confidence Intervals are shown.
(TIF)

**S3 Fig. Serostatus misclassification versus catalytic model estimate bias.** The bias in the estimates from the time-constant and time-varying catalytic models for the realisations over 509 simulated datasets are plotted against the serostatus misclassification error rate. The serostatuses of the titres in each of the 509 simulated datasets are classified as seropositive or seronegative using realisation-specific optimised titre thresholds. The serostatus misclassification error rate is calculated as the percentage of titres in each dataset that are misclassified. Absolute bias is calculated as the absolute value of the estimated value–true value for the force of infection (FOI) and seroprevalence. The linear regression lines and corresponding 95% Confidence Intervals are shown, as well as the Pearson's correlation coefficients (*R*). Three outliers with FOI estimates > 0.4 were removed from the time-varying catalytic model panel and corresponding regression line estimation.
(TIF)

**S4 Fig. Force of infection (FOI) estimates across time.** Yearly FOI estimates from the catalytic and mixture models. 95% Confidence Intervals were calculated by bootstrapping.
(TIF)

## Acknowledgments

We would like to acknowledge Sanofi Pasteur for provision of the data used in this research. We acknowledge the participants, their families and the principal investigators involved in the studies.

## Author Contributions

**Conceptualization:** Victoria Cox, Megan O'Driscoll, Natsuko Imai, Ilaria Dorigatti.

**Data curation:** Ari Prayitno, Sri Rezeki Hadinegoro, Anne-Frieda Taurel, Laurent Coudeville.

**Formal analysis:** Victoria Cox, Megan O'Driscoll.

**Funding acquisition:** Victoria Cox, Ilaria Dorigatti.

**Investigation:** Victoria Cox, Megan O'Driscoll, Natsuko Imai, Ari Prayitno, Sri Rezeki Hadinegoro, Anne-Frieda Taurel, Laurent Coudeville, Ilaria Dorigatti.

**Methodology:** Victoria Cox, Megan O'Driscoll, Natsuko Imai, Ilaria Dorigatti.

**Project administration:** Ilaria Dorigatti.

**Resources:** Ari Prayitno, Sri Rezeki Hadinegoro.

**Software:** Victoria Cox, Megan O'Driscoll.

**Supervision:** Natsuko Imai, Ilaria Dorigatti.

**Validation:** Victoria Cox, Megan O'Driscoll.

**Visualization:** Victoria Cox.

**Writing – original draft:** Victoria Cox, Megan O'Driscoll, Natsuko Imai, Ilaria Dorigatti.

**Writing – review & editing:** Victoria Cox, Megan O'Driscoll, Natsuko Imai, Ari Prayitno, Sri Rezeki Hadinegoro, Anne-Frieda Taurel, Laurent Coudeville, Ilaria Dorigatti.

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
