## [Decision Letter · Decision Letter 0]

1 Dec 2021

Dear Ms Cox,

Thank you very much for submitting your manuscript "Estimating dengue transmission intensity from serological data: a comparative analysis using mixture and catalytic models." for consideration at PLOS Neglected Tropical Diseases. As with all papers reviewed by the journal, your manuscript was reviewed by members of the editorial board and by several independent reviewers. In light of the reviews (below this email), we would like to invite the resubmission of a significantly-revised version that takes into account the reviewers' comments. 

We cannot make any decision about publication until we have seen the revised manuscript and your response to the reviewers' comments. Your revised manuscript is also likely to be sent to reviewers for further evaluation.

Sincerely,

Kendall McKenzie

Staff

Joseph Wu

Deputy Editor

Reviewer's Responses to Questions

**Key Review Criteria Required for Acceptance?**

**Methods**

-Are the objectives of the study clearly articulated with a clear testable hypothesis stated?

-Is the study design appropriate to address the stated objectives?

-Is the population clearly described and appropriate for the hypothesis being tested?

-Is the sample size sufficient to ensure adequate power to address the hypothesis being tested?

-Were correct statistical analysis used to support conclusions?

-Are there concerns about ethical or regulatory requirements being met?

Reviewer #1: (No Response)

Reviewer #2: This paper by Cox et al. compares different methods to estimate the force of infection and the seroprevalence of DENV from cross-sectional seroprevalence surveys. They show that serocatalytic models induce a bias in the parameter estimations compared to a mixture model. I think the authors do a good job at advocating the use of quantitative antibody titers. I appreciated the simulation study, and the statistical analysis is done correctly.

**Results**

-Does the analysis presented match the analysis plan?

-Are the results clearly and completely presented?

-Are the figures (Tables, Images) of sufficient quality for clarity?

Reviewer #1: (No Response)

Reviewer #2: (No Response)

**Conclusions**

-Are the conclusions supported by the data presented?

-Are the limitations of analysis clearly described?

-Do the authors discuss how these data can be helpful to advance our understanding of the topic under study?

-Is public health relevance addressed?

Reviewer #1: (No Response)

Reviewer #2: The authors show well the importance of using quantitative serological data to estimate key epidemiological parameters. I have however some concerns about the novelty of the work. Moreover, some of the points raised in the discussion did not convince me.

**Editorial and Data Presentation Modifications?**

Reviewer #1: (No Response)

Reviewer #2: Minor points

I would suggest minor changes to the figures.

- Figure 1: the y axis need to be consistent, for instance the coverage has to go between 0 and 100. 

- Figure 5, it would be better to have the same x axis in all panels.

- The labels of the supplementary figures are not consistent (they start at Figure S5 in the main text, and at S1 in the supplementary material)

Reference 22: Bollaerts et al., part of the title is missing

**Summary and General Comments**

Reviewer #1: (No Response)

Reviewer #2: 1/ Using mixture models to extract epidemiological parameters from serological studies is quite common now. The authors cite a few papers that use mixture models to estimate the FOI and claim line 97 that they are the first to use these methods to estimate DENV FOI. I’m not yet convinced that applying these methods to estimate DENV FOI is a big improvement to what was done in the various cited papers that used mixture models. Unless accounting for specificities of this virus (like primary/post-primary infections) or for instance specificities in the type of serological data, I don’t see a big novelty brought by the fact that the virus studied here is DENV. It is thus important to show more clearly the novelty of the work in the introduction. 

2/ In the author summary, the authors write that “in high transmission areas catalytic and mixture models produced consistent estimates”. I think this is a bit misleading and hard to understand when compared to Figure 2 that show that the difference between the mixture and catalytic models increase with the FOI. The claim in the summary seems to be not general but true for the Indonesian dataset – which is obtained in a context of high circulation, and where catalytic and mixture models produced consistent estimates – but it seems that the main reason for this consistency is the little overlap in the negative and positive titer distribution. 

If Figure 2 was generated with the values of the seronegative and seropositive components estimated for dataset B (the distributions and values of muS/muI), would they observe a similar difference between the mixture models and catalytic estimates? I expect no impact of the high value of the seroprevalence. 

3/ Line 330 : “Differences in the degree of overlap between components in real serological datasets (S8 Figure) reflect differences in transmission intensity and variable degrees of spatiotemporal heterogeneities in the risk of infection (3,37). “

The titer distributions in the datasets of Vietnam vs Indonesia are so different, that I doubt that they are simply reflecting differences in the exposure to the virus. Since it seems that different ELISA tests were used for datasets A and B, isn’t it the most likely reason for the different titer distributions? 

“In datasets collected from areas which experience hyperendemic DENV circulation, one expects greater separation between the titre components because most seropositive individuals likely have had multiple infections, translating to higher antibody titres”

I don’t believe this is true. If it were true, the way I interpret the Indonesian dataset based on this claim is that some individuals have never been exposed to the virus, whereas the other ones have been exposed multiple times. Is it realistic, and is there a way to check this? I suggest to break down by age groups the titer distribution of the seropositive components. Younger kids are expected to having been fewer infections than older ones. 

4/ The titer values lines 142 and in Figure 3 for dataset B are not consistent. 

5/ lines 138-140: I’d like to see the titer t used as a threshold. For instance as a vertical line in Figure 3.

PLOS authors have the option to publish the peer review history of their article (what does this mean?). If published, this will include your full peer review and any attached files.

Reviewer #1: Yes: Luc E. Coffeng

Reviewer #2: No
---

## [Decision Letter · Decision Letter 1]

11 May 2022

Dear Ms Cox,

Thank you very much for submitting your manuscript "Estimating dengue transmission intensity from serological data: a comparative analysis using mixture and catalytic models." for consideration at PLOS Neglected Tropical Diseases. As with all papers reviewed by the journal, your manuscript was reviewed by members of the editorial board and by several independent reviewers. The reviewers appreciated the attention to an important topic. Based on the reviews, we are likely to accept this manuscript for publication, providing that you modify the manuscript according to the review recommendations. 

Sincerely,

Joseph T. Wu

Deputy Editor

Joseph Wu

Deputy Editor

Reviewer's Responses to Questions

**Key Review Criteria Required for Acceptance?**

**Methods**

-Are the objectives of the study clearly articulated with a clear testable hypothesis stated?

-Is the study design appropriate to address the stated objectives?

-Is the population clearly described and appropriate for the hypothesis being tested?

-Is the sample size sufficient to ensure adequate power to address the hypothesis being tested?

-Were correct statistical analysis used to support conclusions?

-Are there concerns about ethical or regulatory requirements being met?

Reviewer #1: (No Response)

Reviewer #2: No additional comments.

**Results**

-Does the analysis presented match the analysis plan?

-Are the results clearly and completely presented?

-Are the figures (Tables, Images) of sufficient quality for clarity?

Reviewer #1: (No Response)

Reviewer #2: No additional comments.

**Conclusions**

-Are the conclusions supported by the data presented?

-Are the limitations of analysis clearly described?

-Do the authors discuss how these data can be helpful to advance our understanding of the topic under study?

-Is public health relevance addressed?

Reviewer #1: (No Response)

Reviewer #2: No additional comments.

**Editorial and Data Presentation Modifications?**

Reviewer #1: (No Response)

Reviewer #2: In this revision, Cox et al. respond adequately to the requests of the reviewers and I found the article much clearer in this version.

1/ I have one comment about the equations in the Methods section: they are part of the text and the punctuation should therefore be made accordingly. 

2/ Are the datasets publicly available? This seems to be true for the Vietnam dataset and not for the Indonesian one. Please give more details about data availability.

**Summary and General Comments**

Reviewer #1: The author's efforts to address the reviewer comments are appreciated, in particular the clarification on what the models' underlying assumptions and what they are supposed to represent. This clarification has, however, raised a concern about the appropriateness of how the catalytic model was implemented in terms of maths. Equation 1 seems incorrect, or at least does not match the assumptions and model purpose described in the introduction. For details, please see the attachment.

Reviewer #2: No additional comments.

PLOS authors have the option to publish the peer review history of their article (what does this mean?). If published, this will include your full peer review and any attached files.

Reviewer #1: Yes: Luc E. Coffeng

Reviewer #2: No

Figure Files:

Data Requirements:

Reproducibility:

References

---

## [Decision Letter · Decision Letter 2]

16 Jun 2022

Dear Ms Cox,

We are pleased to inform you that your manuscript 'Estimating dengue transmission intensity from serological data: a comparative analysis using mixture and catalytic models.' has been provisionally accepted for publication in PLOS Neglected Tropical Diseases.

Best regards,

Joseph T. Wu

Deputy Editor

Joseph Wu

Deputy Editor

Reviewer's Responses to Questions

**Key Review Criteria Required for Acceptance?**

**Methods**

-Are the objectives of the study clearly articulated with a clear testable hypothesis stated?

-Is the study design appropriate to address the stated objectives?

-Is the population clearly described and appropriate for the hypothesis being tested?

-Is the sample size sufficient to ensure adequate power to address the hypothesis being tested?

-Were correct statistical analysis used to support conclusions?

-Are there concerns about ethical or regulatory requirements being met?

Reviewer #1: (No Response)

Reviewer #2: (No Response)

**Results**

-Does the analysis presented match the analysis plan?

-Are the results clearly and completely presented?

-Are the figures (Tables, Images) of sufficient quality for clarity?

Reviewer #1: (No Response)

Reviewer #2: (No Response)

**Conclusions**

-Are the conclusions supported by the data presented?

-Are the limitations of analysis clearly described?

-Do the authors discuss how these data can be helpful to advance our understanding of the topic under study?

-Is public health relevance addressed?

Reviewer #1: (No Response)

Reviewer #2: (No Response)

**Editorial and Data Presentation Modifications?**

Reviewer #1: (No Response)

Reviewer #2: (No Response)

**Summary and General Comments**

Reviewer #1: (No Response)

Reviewer #2: The authors answered the questions of the reviewers and I find no additional comment to add to my review. I therefore recommend this paper for publication.

PLOS authors have the option to publish the peer review history of their article (what does this mean?). If published, this will include your full peer review and any attached files.

Reviewer #1: **Yes: **Luc E. Coffeng

Reviewer #2: No

---

## [Editor Report · Acceptance letter]

6 Jul 2022

Dear Ms Cox,

We are delighted to inform you that your manuscript, "Estimating dengue transmission intensity from serological data: a comparative analysis using mixture and catalytic models.," has been formally accepted for publication in PLOS Neglected Tropical Diseases.

Best regards,

Shaden Kamhawi

co-Editor-in-Chief

Paul Brindley

co-Editor-in-Chief
